# Imaging neuropeptide release at synapses with a genetically engineered reporter

Keke Ding[1†], Yifu Han[2,3†], Taylor W Seid[1], Christopher Buser[4], Tomomi Karigo[1], Shishuo Zhang[1], Dion K Dickman[2], David J Anderson[1,5,6]*

[1]Division of Biology and Biological Engineering, California Institute of Technology, Pasadena, United States; [2]Department of Neurobiology, University of Southern California, Los Angeles, United States; [3]Neuroscience Graduate Program, University of Southern California, Los Angeles, United States; [4]Oak Crest Institute of Science, Monrovia, United States; [5]Howard Hughes Medical Institute, California Institute of Technology, Pasadena, United States; [6]Tianqiao and Chrissy Chen Institute for Neuroscience, California Institute of Technology, Pasadena, United States

**Abstract** Research on neuropeptide function has advanced rapidly, yet there is still no spatio-temporally resolved method to measure the release of neuropeptides in vivo. Here we introduce Neuropeptide Release Reporters (NPRRs): novel genetically-encoded sensors with high temporal resolution and genetic specificity. Using the *Drosophila* larval neuromuscular junction (NMJ) as a model, we provide evidence that NPRRs recapitulate the trafficking and packaging of native neuropeptides, and report stimulation-evoked neuropeptide release events as real-time changes in fluorescence intensity, with sub-second temporal resolution.

**\*For correspondence:**
wuwei@caltech.edu

[†]These authors contributed equally to this work

**Competing interests:** The authors declare that no competing interests exist.

## Introduction

Neuropeptides (NPs) exert an important but complex influence on neural function and behavior (*Hökfelt et al., 2000*; *Insel and Young, 2000*; *Nässel and Winther, 2010*; *Bargmann and Marder, 2013*). A major lacuna in the study of NPs is the lack of a method for imaging NP release in vivo, with subcellular spatial resolution and subsecond temporal resolution. Available techniques for measuring NP release include microdialysis (*Kendrick, 1990*), antibody-coated microprobes (*Schaible et al., 1990*) and GFP-tagged propeptides visualized either by standard fluorescence microscopy (*van den Pol, 2012*), or by TIRF imaging of cultured neurons (*Xia et al., 2009*). In *Drosophila*, a fusion between rat Atrial Natriuretic Peptide/Factor (ANP/F) and GFP was used to investigate neuropeptide trafficking at the fly neuromuscular junction (NMJ) (*Rao et al., 2001*). Release was measured indirectly, as a decrease in ANP-GFP fluorescence intensity at nerve terminals reporting residual unreleased peptide, on a time-scale of seconds (*Wong et al., 2015*). None of these methods combines NP specificity, genetically addressable cell type-specificity, high temporal resolution and applicability to in vivo preparations (*Supplementary file 1*). A major challenge is to develop a tool that encompasses all these features for direct, robust measurement of NP release in vivo.

## Results

Neuropeptides are synthesized as precursors, sorted into dense core vesicles (DCVs), post-translationally modified and cleaved into active forms prior to release (*Taghert and Veenstra, 2003*). We reasoned that an optimal in vivo real-time NP release reporter should include (1) a reporter domain that reflects the physico-chemical contrast between the intravesicular milieu and the extracellular space (*Figure 1—figure supplement 1A*); and (2) a sorting domain that ensures its selective trafficking into DCVs (*Figure 1—figure supplement 1b*). The NP precursor may function as the sorting

domain, suggested by studies of DCV fusion using pIAPP-EGFP (*Barg et al., 2002*) and NPY-pHluorin (*Zhu et al., 2007*) in cultured neurons, or ANP-GFP in *Drosophila* (*Rao et al., 2001*). We therefore developed a pipeline to screen various transgenes comprising NP precursors fused at different sites to fluorescent reporters, in adult flies (*Figure 1—figure supplement 1B–C*). A total of 54 constructs were tested. We found that optimal trafficking was achieved by substituting the reporter for the NP precursor C-terminal domain that follows the final peptide (*Figure 1—figure supplement 1B*). In order to maintain covalent linkage with the reporter domain, we removed the dibasic cleavage site C-terminal to the final peptide.

The DCV lumen has lower pH and free calcium (pH = 5.5–6.75, $[Ca^{2+}]{\sim}30$ μM) compared to the extracellular space (pH = 7.3, $[Ca^{2+}]{\sim}2$ mM) (*Mitchell et al., 2001*; *Sturman et al., 2006*). These differences prompted us to test validated sorting domains in a functional ex vivo screen using either pH-sensitive fluorescent proteins (*Miesenböck et al., 1998*) or genetically-encoded calcium indicators (GECIs) (*Tian et al., 2012*; *Lin and Schnitzer, 2016*) (*Figure 1—figure supplement 1A–D*). Reporters based on pHluorins (*Miesenböck et al., 1998*) did not perform well in our hands, therefore we focused on GCaMP6s (*Chen et al., 2013*). The calcium sensitivity threshold of GCaMP6s is below the calcium concentration in both DCVs and the extracellular space. However, GCaMP6s fluorescence is quenched in the acidic DCV lumen (*Barykina et al., 2016*), enabling it to function as a dual calcium/pH indicator (*Figure 1A*). These key properties should boost the contrast between GCaMP6s fluorescence in unreleased vs. released DCVs, potentially allowing us to trace NP release at the cellular level in vivo.

We sought to test several NP precursor-GCaMP6s fusion proteins, called NPRRs (**N**euro**P**eptide **R**elease **R**eporters; unless otherwise indicated all NPRRs refer to fusions with GCaMP6s), in an intact preparation using electrical stimulation to evoke release. Initially for proof-of-principle experiments, we used the Dro*sophila* larval NMJ to test NPRR$^{ANP}$, a GCaMP6s fusion with rat ANP (*Burke et al., 1997*). NMJ terminals are large, individually identifiable, and easy to image and record. In particular, boutons on muscle 12/13 are diverse – Type Ib and Type Is boutons contain mostly synaptic vesicles and few DCVs, while Type III boutons contain an abundance of DCVs but no synaptic vesicles (*Menon et al., 2013*); moreover, Type III-specific GAL4 drivers are available (*Koon and Budnik, 2012*) (*Figure 1B*).

Expression of NPRR$^{ANP}$ pan-neuronally (under the control of nsyb-GAL4) followed by double immuno-staining for ANP and GCaMP (anti-GFP) indicated that the sorting domain and the reporter domains showed a similar localization in Type III neurons (*Figure 1—figure supplement 2*). Moreover, the distribution of NPRR$^{ANP}$ overlapped that of Bursicon (*Figure 1—figure supplement 3D*), an NP that is endogenously expressed in Type III neurons (*Loveall and Deitcher, 2010*). Both GCaMP and Bursicon immunoreactivity were strongest within boutons, consistent with the known subcellular localization of DCVs (*Gorczyca and Budnik, 2006*).

Glutamate is the only known canonical neurotransmitter used at the larval NMJ (*Menon et al., 2013*). This allowed visualization of the subcellular localization of small synaptic vesicles (SV) by immuno-staining for vGluT, a vesicular glutamate transporter (*Fremeau et al., 2001*; *Kempf et al., 2013*). In Type Ib neurons (which contain relatively few DCVs relative to SVs [*Menon et al., 2013*]), vGluT staining was observed as patches with a dim center, which may reflect clustered SVs, while NPRR$^{ANP}$ immunoreactivity was seen in dispersed, non-overlapping punctae (*Figure 1C*, α-GFP, inset). In Type III neurons, NPRRs were strongly expressed but no vGluT immunoreactivity was detected (*Figure 1C*). The subcellular distribution of this NPRR in larval NMJ neurons, therefore, is similar to that of other DCV-targeted markers previously used in this system (*Rao et al., 2001*; *Shakiryanova et al., 2006*), and appears to reflect exclusion from SVs.

The diffraction limit of light microscopy precluded definitive co-localization of NPRRs in DCVs. Therefore, we employed Immuno-Electron microscopy (Immuno-EM) to investigate the subcellular localization of NPRRs at the nanometer scale. To maximize antigenicity for Immuno-EM, we generated constructs that replaced GCaMP6s with GFP (NPRR$^{ANP-GFP}$;). NPRR$^{ANP-GFP}$ showed dense labeling in association with DCVs (*Figure 1D*, arrows), where the average number of gold particles/μm$^2$ was substantially and significantly higher than in neighboring bouton cytoplasm (DCV/Bouton ∼ 14.26) (*Figure 1E*, *Supplementary file 2*). Taken together, these data indicate that NPRR$^{ANP-GFP}$ is localized to DCVs. By extension, they suggest that NPRR$^{ANP-GCaMP6s}$ (which has an identical structure to NPRR$^{ANP-GFP}$ except for the modifications that confer calcium sensitivity) is similarly packaged in DCVs. While these two reporters show indistinguishable distributions by

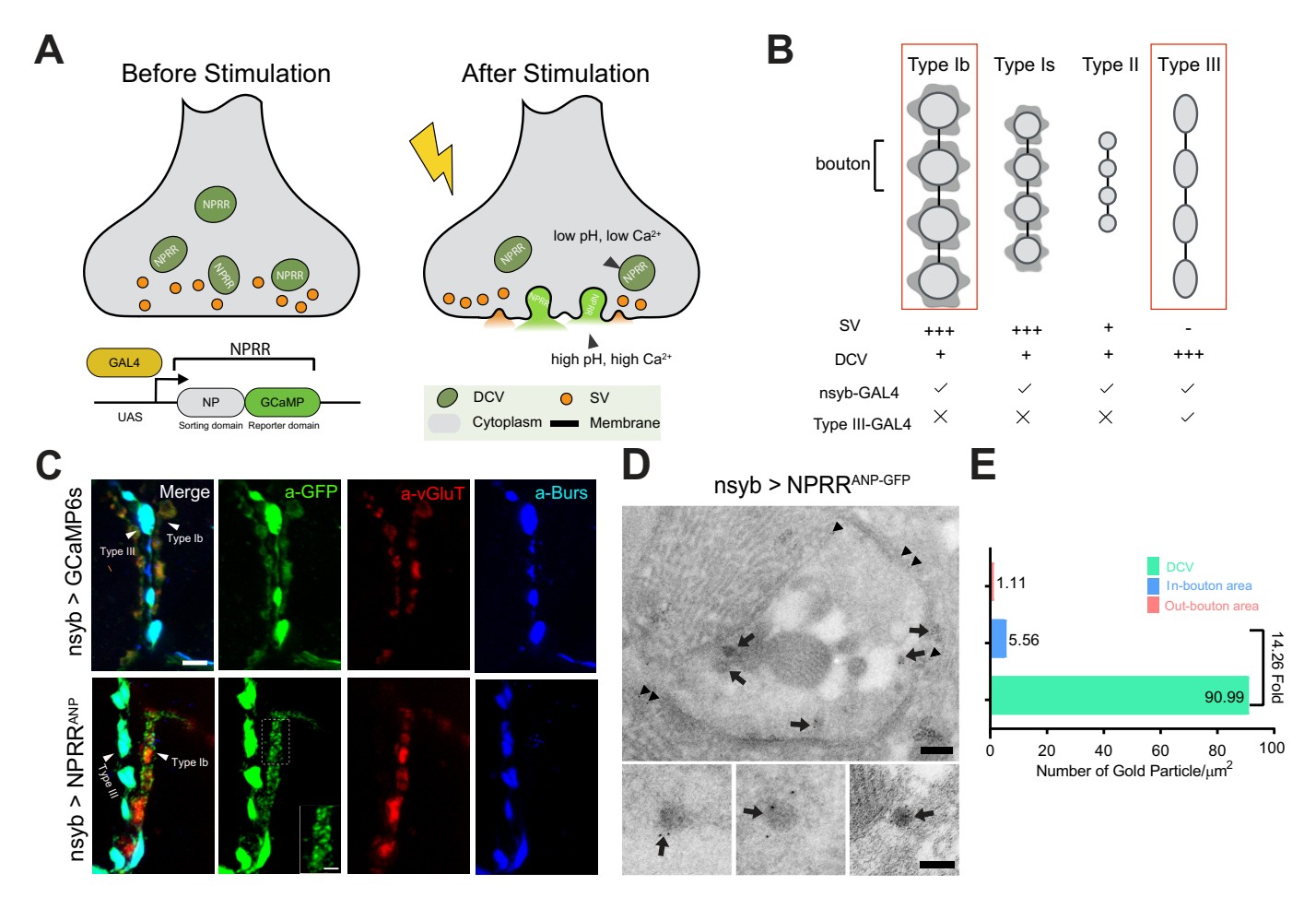

**Figure 1.** Design and Synaptic Localization of an NPRR. (**A**) Schematic illustrating the principle of NPRRs (Neuropeptide Release Reporters). NPRR molecules in the DCV lumen (low pH/low calcium, *left*) exhibit increased fluorescence when released by fusion into the extracellular space (neutral pH/ high calcium, *right*). NPRR fluorescent signal is expected to decay following diffusion into the synaptic cleft. New NPRR-containing DCVs are produced by synthesis and transport from the soma, not by recycling. NP: Neuropeptide. DCV: Dense Core Vesicle. SV: Synaptic Vesicle. (**B**) Distinct motor neuron subtypes at the *Drosophila* NMJ (muscle 12/13) have different proportions of DCVs vs. SVs. The GAL4 driver R57C10-Gal4 (nsyb-GAL4) labels all subtypes, while R20C11-GAL4 selectively labels only Type III neurons, which lack SVs ('Type III-GAL4'). Light gray circles, black lines and dark gray shading represent boutons, inter-bouton intervals and subsynaptic reticulum respectively. The studies in this paper focus on Type Ib neurons and Type III neurons (in red rectangles). (**C**) Triple immunolabeling for GFP (green), Bursicon (blue) and vGluT (red), in flies containing nsyb-GAL4 driving UAS-GCaMP6s (*upper*), or NPRR[ANP] (*lower*). Type Ib and Type III boutons are indicated. Scale bar, 5 μm. Inset image (NPRR[ANP], a-GFP channel) shows details of puncta distribution of NPRR[ANP] in Type Ib neuron. Scale bar, 2 μm. (**D**) TEM images of boutons immunolabeled with anti-GFP (5 nm gold particle-conjugated) to detect nsyb>NPRR[ANP-GFP], which has an identical structure to NPRR[ANP], but is a GFP rather than GCaMP6s fusion to improve antigenicity (see *Figure 1—figure supplement 4*). Note strong labeling in DCVs (arrows) and the neuronal plasma membrane (arrowheads). Scale bar, 200 nm. Lower panel shows representative images of labeled DCVs. Scale bar,100 nm. (**E**) Quantification for TEM images in (**D**).

The online version of this article includes the following source data and figure supplement(s) for figure 1:

**Source data 1.** Raw data for Immuno-EM experiments (ANP).

**Source data 2.** Raw data for Immunno-EM experiments (control group).

**Figure supplement 1.** NPRR screening pipeline.

**Figure supplement 2.** Exogeneous neuropeptide ANP dictates the expression pattern of NPRR[ANP].

**Figure supplement 3.** Expression of different reporters in Type III neurons in the larval NMJ.

**Figure supplement 4.** Subcellular distribution of NPRR[ANP] and NPRR[ANP-GFP].

immunofluorescence (*Figure 1—figure supplement 4*), we cannot formally exclude that the substitution of GCaMP for GFP may subtly alter subcellular localization of the NPRR in a manner undetectable by light microscopy.

To measure the release of NPRRs from DCVs, we next expressed NPRR[ANP] in Type III neurons using a specific GAL4 driver for these cells (*Koon and Budnik, 2012*) (*Figure 2E* and *Figure 1—figure supplement 3D*). We delivered 4 trials of 70 Hz electrical stimulation to the nerve bundle, a frequency reported to trigger NP release as measured by ANF-GFP fluorescence decrease (*Rao et al., 2001*; *Shakiryanova et al., 2006*), and used an extracellular calcium concentration that promotes full fusion mode (*Alés et al., 1999*). This stimulation paradigm produced a relative increase in NPRR[ANP] fluorescence intensity (ΔF/F), whose peak magnitude increased across successive trials (*Figure 2A*, red bars and 2D; *Video 1*; *Figure 2—figure supplement 1*, $A_1$ vs. $A_7$). Responses in each trial showed a tri-phasic temporal pattern: (1) In the 'rising' phase, NPRR[ANP] ΔF/F peaked 0.5–5 secs after stimulation onset, in contrast to the virtually instantaneous peak seen in positive control specimens expressing conventional GCaMP6s in Type III neurons (*Figure 2A–B*). The NPRR[ANP] latency to peak was similar to the reported DCV fusion latency following depolarization in hippocampal neurons (*Xia et al., 2009*). This delay is thought to reflect the kinetic difference between calcium influx and DCV exocytosis due to the loose association between DCVs and calcium channels (*Xia et al., 2009*). (2) In the 'falling' phase, NPRR[ANP] ΔF/F began to decline 1–5 s before the termination of each stimulation trial, presumably reflecting depletion of the available pool of releasable vesicles. In contrast, GCaMP6s fluorescence did not return to baseline until after stimulation offset

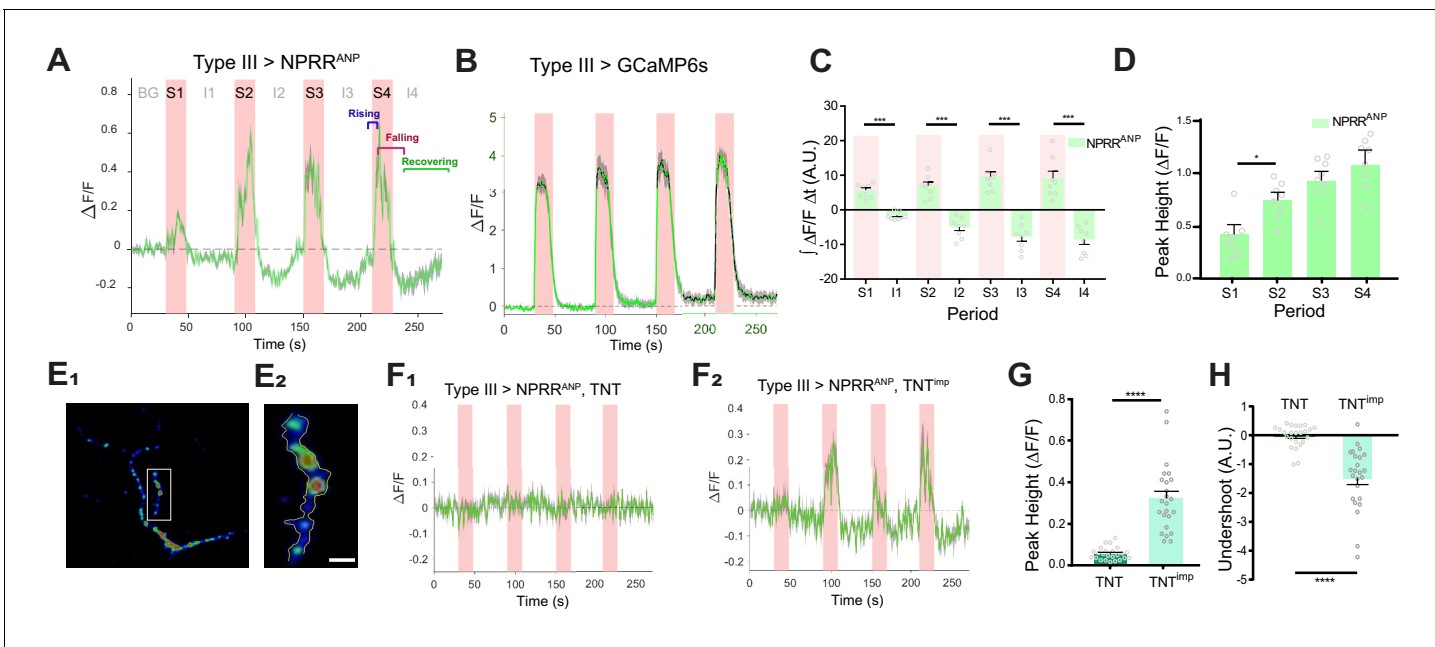

**Figure 2.** NPRR specifically reports neuropeptide release. (**A**) Trace from a representative experiment showing changes in NPRR[ANP] fluorescence intensity (ΔF/F) in Type III motor neurons at the larval NMJ evoked by electrical stimulation. BG: background. S1-S4: Stimulation trials 1–4. I1-I4: Inter-stimulation Intervals (ISIs) 1–4. Green line: ΔF/F averaged across all boutons in the field of view. Gray shading: s.e.m envelope. Red bar: electrical stimulation trials (70 Hz). The three typical phases of the response are indicated in S4. The peak height of the response on the first trial is characteristically lower (see also (**D**)), and may reflect competition with unlabeled DCVs in the readily releasable pool. (**B**) ΔF/F traces in control flies expressing cytoplasmic GCaMP6s in Type III neurons. (**C**) Integrated NPRR[ANP] ΔF/F values during trials S1-4 and intervals I1-4. A.U.: arbitrary units. $n$ = 8. \*\*\*, p<0.001. (**D**) Average NPRR[ANP] ΔF/F peak heights for trials S1-4. $n$ = 8. \*, p<0.05. Plotted values in (**C–D**) are mean ± s.e.m. (**E1–E2**) Representative selection of ROIs (yellow). Details see Materials and methods. Scale bar, 5 μm. (**F**) NPRR[ANP] ΔF/F response are abolished in Type III GAL4>UAS NPRR[ANP] flies bearing UAS-TNT (**F1**) but not UAS-TNT[imp] (**F2**). (**G**) Average peak heights of NPRR[ANP] ΔF/F in combined stimulation trials (S1-4) from (**F**). \*\*\*\*, p<0.0001. (**H**) Average 'undershoot', defined as the integrated ΔF/F during ISIs I1-4 (see (**C**)). In (**C–D**) and (**G–H**).

The online version of this article includes the following figure supplement(s) for figure 2:

**Figure supplement 1.** Activation of NPRR[ANP]in situ.

**Figure supplement 2.** NPRR specifically reports neuropeptide release.

**Figure supplement 3.** Blocking DCV fusion using Tetanus Toxin.

(*Figure 2A–B*). (3) Finally, unlike GCaMP6s, NPRR[ANP] exhibited an 'undershoot' (ΔF/F below baseline) during the post-stimulation intervals, followed by a 'recovering' phase (*Figure 2A*; *Figures 2C,I1–4*). This undershoot may reflect dilution of released fluorescent NPRR molecules

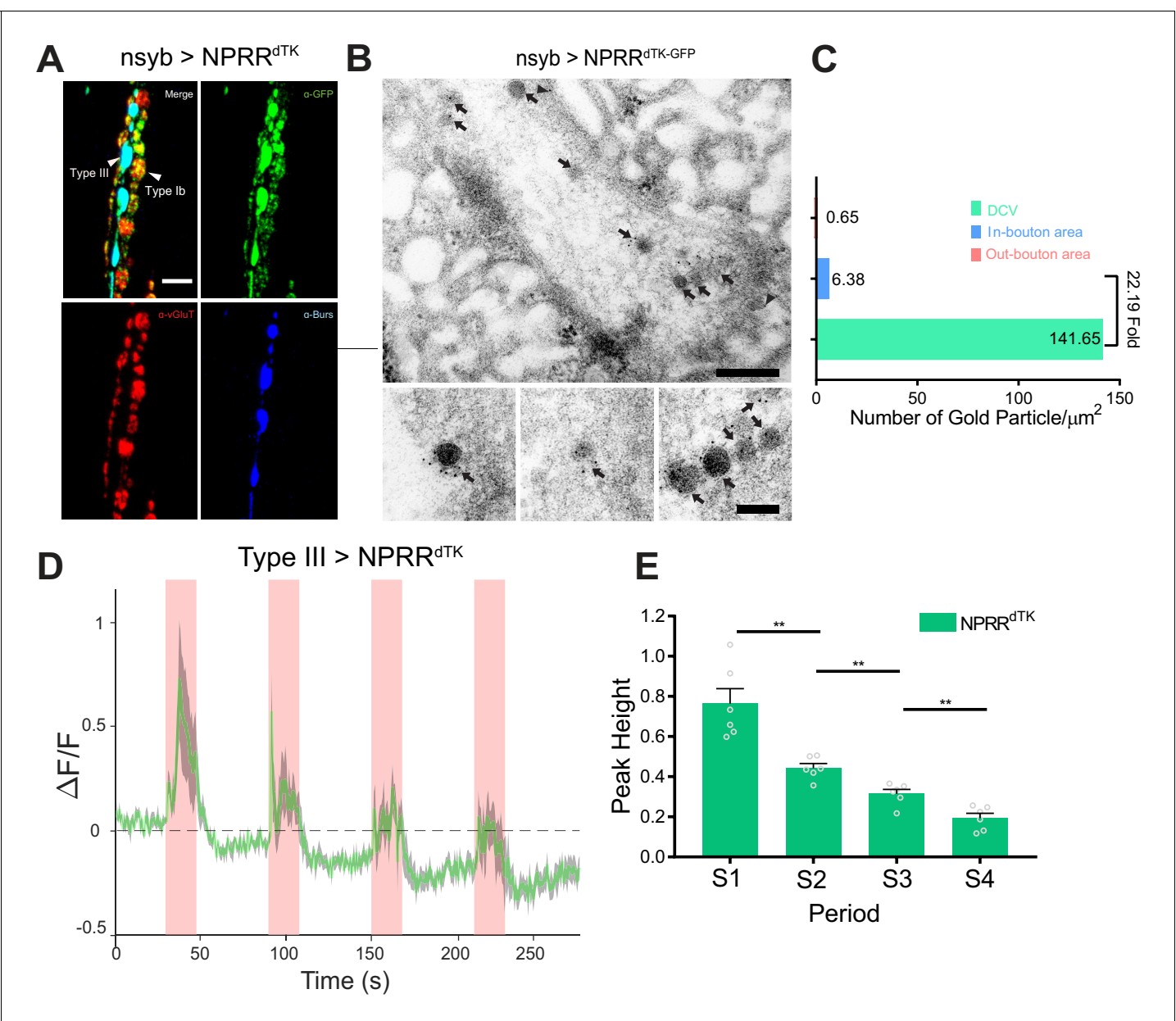

**Figure 3.** Application of the NPRR approach to a *Drosophila* neuropeptide. (**A**) Triple immunolabeling for GFP (green), Bursicon (blue) and vGluT (red) in Type III-GAL4> UAS NPRR[dTK] flies. Scale bar, 5 μm. (**B**) TEM images of boutons immunolabeled against GFP (5 nm gold) in nsyb-GAL4>UAS NPRR[dTK-GFP] flies. Note strong labeling in DCVs (arrows) and bouton plasma membrane (arrowheads). Scale bar, 200 nm. Lower panel shows representative images of labeled DCVs. Scale bar,100 nm. (**C**) Quantification of TEM images in (**B**). (**D**) NPRR[dTK] ΔF/F curve; stimulation conditions as in *Figure 2A*. (**E**) Average NPRR[dTK] ΔF/F peak height above pre-stimulation baseline (corrected; see Materials and methods) for stimulation trials S1-4. $n = 6$. \*\*, $p<0.01$.

The online version of this article includes the following source data for figure 3:

**Source data 1.** Raw data for Immuno-EM experiments (dTK).

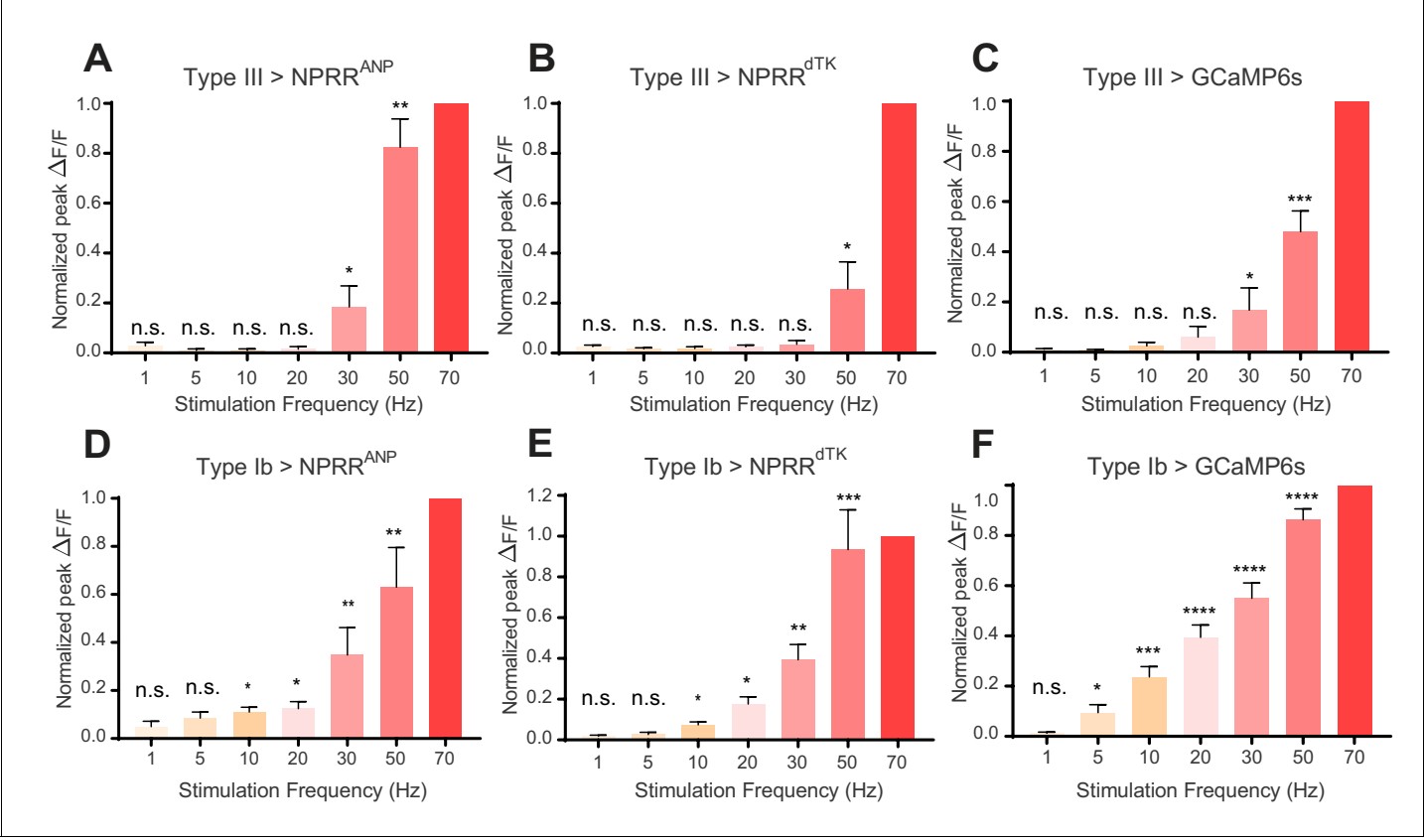

**Figure 4.** NPRR reveals distinct cell-type specific peptide release properties. For each preparation, a series of stimulation trials were delivered at frequencies from 1 Hz to 70 Hz, as indicated. In-stimulation response peaks were normalized to 70 Hz. The normalized peaks of NPRRs or calcium responses (measured with cytosolic GCaMP6s) were pooled and plotted for both Type III (A-C) and Type Ib (D-F) neurons. Responses were compared to zero. n = 6–12. n.s., not significant. *, p<0.05. **, p<0.01. ***, p<0.001. ****, p<0.0001.

The online version of this article includes the following figure supplement(s) for figure 4:

**Figure supplement 1.** Comparison of NPRR response at 30 and 50 Hz.

by diffusion into the synaptic cleft (*van den Pol, 2012*), while recovery may reflect DCV replenishment in the boutons from vesicles proximal to the imaged release site.

Because NPRR$^{ANP}$ fluorescence was preferentially accumulated within boutons, we asked whether these regions contributed to ΔF/F peaks more significantly than the inter-bouton intervals (IBIs). To do this, we partitioned the processes into boutons and IBI fields (*Figure 2—figure supplement 2A*), and compared the ΔF/F in these regions during stimulation trials. The time-averaged ratio of bouton/IBI ΔF/F (see Materials and Methods) was significantly higher for NPRR$^{ANP}$ than for GCaMP6s, particularly during later stimulation trials (*Figure 2—figure supplement 2B*, green bars, S2-4). This contrast indicates that NPRR$^{ANP}$ signals are preferentially observed in boutons, where DCVs are located, and do not reflect differences in cytoplasmic free Ca$^{2+}$ levels between these regions as detected by GCaMP6s.

To test definitively if NPRR$^{ANP}$ ΔF/F signals are dependent upon NP release, we blocked vesicle fusion at terminals of Type III neurons using expression of tetanus toxin light chain (TNT) (*Sweeney et al., 1995*), a protease that cleaves n-synaptobrevin, a v-snare required for DCV fusion (*Figure 2—figure supplement 3*) (*Xu et al., 1998*). As a control, we used impotent TNT (TNT$^{imp}$), a reduced activity variant (*Sweeney et al., 1995*). TNT expression completely abolished stimulation-induced ΔF/F increases from NPRR$^{ANP}$, while TNT$^{imp}$ did not (*Figure 2F*). Further analysis revealed that both the ΔF/F peaks and inter-stimulation undershoots were diminished by TNT (*Figure 2G–H*). In contrast, neither TNT nor TNT$^{imp}$ affected the kinetics of GCaMP6s signals in Type III neurons

(*Figure 2—figure supplement 2C*), which report cytosolic $Ca^{2+}$ influx. Taken together, these data support the idea that NPRR$^{ANP}$ signals specifically reflect DCV release.

ANP is a rat NP that lacks a *Drosophila* homolog (*Rao et al., 2001*). To determine whether our method could be applied to detect the release of a specific, endogenous fly NP, we tested NPRR$^{dTK}$, one of 6 different reporter variants we initially generated from the *Drosophila* neuropeptide precursor, DTK (*Figure 1—figure supplement 1B*). In contrast to ANP which encodes a single peptide, DTK yields multiple NP derivatives (*Winther et al., 2003*). Light microscopy (*Figure 3A*) and Immuno-EM (*Figure 3B*, arrows) confirmed that NPRR$^{dTK}$, like NPRR$^{ANP}$, was localized to DCVs (DCV/bouton ~ 22.19, *Figure 3C*). Using the Type III-specific GAL4 driver to express NPRR$^{dTK}$ and the same stimulation protocol as used for NPRR$^{ANP}$, the basic tri-phasic response profile was also observed (*Figure 3D*). However, peak heights and baseline fluorescence fell progressively with successive stimulation trials (*Figure 3E*), in contrast to NPRR$^{ANP}$ where the first peak and undershoot were lower (*Figure 2C–D*). The reason for this difference is currently unclear.

We next investigated the relationship between NPRR signal and stimulation intensity, by delivering to the Type III neurons a series of low to high frequency electrical stimuli (1–70 Hz; *Levitan et al., 2007*) while imaging the nerve terminals. For direct comparison of NPRR responses across different preparations, we applied a posteriori normalization of fluorescent peaks in each trial to the highest response obtained among all trials. For both NPRR$^{ANP}$ and NPRR$^{dTK}$ (*Figure 4A–B*), the peak responses showed a positive correlation with stimulation frequency, analogous to that observed using cytosolic GCaMP6s (*Figure 4C*). In Type III neurons, the responses of both NPRRs to stimulation frequencies < 30 Hz (1,5,10,20 Hz) were not statistically significant from zero. NPRR$^{ANP}$ showed a higher sensitivity to high stimulation frequencies (30 Hz: 18.14%, 50 Hz: 82.40% Normalized peak $\Delta F/F$), while NPRR$^{dTK}$ showed a higher stimulation threshold and lower sensitivity (30 Hz: 3.57%, 50 Hz: 24.67% Normalized peak $\Delta F/F$).

We next investigated whether the relatively high stimulation frequency required to observe significant responses with NPRRs was a function of the reporters, or rather of the cell class in which they were tested. To do this, we expressed both NPRRs in Type Ib neurons, a class of motor neurons that contains both SVs and DCVs (*Figure 1B*, *Figure 4D–F*), and performed stimulation frequency titration experiments. Strikingly, in Type Ib neurons, significant increases in $\Delta F/F$ could be observed at frequencies as low as 10 Hz (*Figure 4D,E*; NPRR$^{ANP}$ @ 20 Hz: 12.50%, NPRR$^{dTK}$ @ 20 Hz: 17.67% normalized peak $\Delta F/F$). The reason for the difference in NPRR threshold between Type III and Type Ib neurons is unknown, but parallels their difference in GCaMP6s response to electrical stimulation (*Figure 4C* vs. *Figure 4F*).

Notably, although NPRR$^{ANP}$ and NPRR$^{dTK}$ presented distinct response profiles in Type III neurons, their performance in Type Ib neurons was more similar (*Figure 4A* vs. *Figure 4B*; cf. *Figure 4D* vs. *Figure 4E*). In summary, the differences in performance we observed between the two NPRRs appeared to be specific to Type III neurons, and were minor in comparison to the differences in performance of both reporters between the two cell classes. The reason for the differences between NPRR$^{ANP}$ and NPRR$^{dTK}$ sensitivity and kinetics in Type III neurons is unknown but may reflect differences in how well these reporters compete with the high levels of endogenous neuropeptide (Bursicon) for packaging, transport or release.

## Discussion

Here we present proof-of-principle for a method to detect the release of different neuropeptides in intact neural tissue, with subcellular spatial and sub-second temporal resolution. By exploiting the fluorescent change of GCaMP in response to a shift in pH and $[Ca^{2+}]$, we visualized the release of neuropeptides by capturing the difference between the intravesicular and extracellular microenvironment. NPRR responses exhibited triphasic kinetics, including rising, falling and recovering phases. In the falling phase, a post-stimulus 'undershoot', was observed in which the fluorescent intensity fell below pre-stimulation baseline. This undershoot presumably reflect the slow kinetics of DCV replenishment relative to release.

The molecular mechanisms of NP release are incompletely understood (*Xu and Xu, 2008*). It is possible that individual DCVs only unload part of their cargo during stimulation, in which case many DCVs that underwent fusion may still contain unreleased NPRR molecules following a stimulus pulse. Although we are convinced that NPRR signals do indeed reflect NP release, due to the presence of

the recovering phase, we cannot formally exclude that unreleased NPRRs may contribute to the signal change due to their experience of intravesicular $[Ca^{2+}]$/pH changes that occur during stimulation. To resolve this issue in the future, an ideal experiment would be to co-express an NPRR together with a $[Ca^{2+}]$/pH-invariant NP-reporter fusion. Multiple attempts to generate such fusions with RFP were unsuccessful, due to cryptic proteolytic cleavage sites in the protein which presumably result in degradation by DCV proteases during packaging.

To test if NPRR$^{ANP}$ ΔF/F signals are dependent on NP release, we expressed the light chain of tetanus toxin (TNT), a reagent shown to effectively block NP release in many (*McNabb and Truman, 2008*; *Hentze et al., 2015*; *Zandawala et al., 2018*), if not all (*Umezaki et al., 2011*), systems. We observed a striking difference in NPRR kinetics in flies co-expressing TNT vs. its proteolytically inactive 'impotent' control form TNT$^{imp}$ (*Figure 2F*). The strong reduction of NPRR signals by TNT-mediated n-syb cleavage is consistent with the idea that these signals reflect the release of NPRRs from DCVs.

We have tested the generalizability of the principles used to generate NPRRs by (1) constructing a surrogate NP reporter NPRR$^{ANP}$ as well as a multi-peptide-producing endogenous *Drosophila* NP reporter NPRR$^{dTK}$ (*Figures 2–3*); (2) characterized NPRR signals in response to varying intensities of electrical stimulation; and (3) recorded NPRR signals in two different classes of NMJ motor neurons containing DCVs with or without SVs, respectively (*Figure 4*). These experiments revealed, to our surprise, that NPRR responses exhibit cell-type specific characteristics (*Figure 4*). As NPRRs are applied to other neuropeptides and cell types, a systematic characterization of neuropeptide release properties in different peptidergic neurons should become possible, furthering our understanding of neuropeptide biology.

The method described here can, in principle, be extended to an in vivo setting. This would open the possibility of addressing several important unresolved issues in the study of NP function in vivo. These include the 'which' problem (which neuron(s) release(s) NPs under particular behavioral conditions?); the 'when' problem (when do these neurons release NPs relative to a particular behavior or physiological event?); the 'where' problem (are NPs released from axons, dendrites or both?); and the 'how' problem (how is NP release regulated?). The application of NPRRs to measuring NP release dynamics in awake, freely behaving animals may yield answers to these important long-standing questions.

# Materials and methods

## Key resources table

| Reagent type (species) or resource | Designation | Source or reference | Identifiers | Additional information |
|---|---|---|---|---|
| Genetic reagent (*D. melanogaster*) | UAS-NPRR$^{ANP}$ (attp2) | this paper | | See Materials and methods, subsection Construction of transgenic animals. |
| Genetic reagent (*D. melanogaster*) | UAS-NPRR$^{dTK}$ (attp2) | this paper | | Same as above. |
| Genetic reagent (*D. melanogaster*) | UAS-TNT$^{imp}$ | Bloomington Drosophila Stock Center | BDSC:28840; FLYB:FBti0038575; RRID:BDSC_28840 | Flybase symbol: w[*]; P{w[+mC]=UAS TeTxLC.(-)V}A2 |
| Genetic reagent (*D. melanogaster*) | UAS-TNT | Bloomington Drosophila Stock Center | BDSC:28838; FLYB:FBti0038527; RRID:BDSC_28838 | Flybase symbol: w[*]; P{w[+mC]=UAS TeTxLC.tnt}G2 |
| Genetic reagent (*D. melanogaster*) | w; +; UAS-GCaMP6s (su(Hw)attp1) | *Hoopfer et al., 2015* | | |
| Antibody | anti-GFP (chicken polyclonal) | Aveslab | Aveslab: GFP-1020; RRID:AB_2307313 | (1:250:Immuno-EM, 1:1000: IHC) |
| Antibody | anti-ANP (rabbit polyclonal) | abcam | abcam #14348 | (1:500) |

## Fly strains

All experimental flies were reared on a 12/12 hr day-night cycle at 25°C. Standard chromosomal balancers and genetic strategies were used for all crosses and for maintaining mutant lines. Detailed genotypes used are summarized in *Supplementary file 3*. The following strains were obtained from Bloomington Stock Center (Indiana University): R20C11-Gal4 (#48887), R57C10-Gal4 (#39171), UAS-mCD8::GFP (#32185), UAS-TNT (#28838), UAS-TNT$^{imp}$ (#28840). UAS-opGCaMP6s was made by Barret Pfeiffer (Gerald Rubin's lab, Janelia Farm) (*Hoopfer et al., 2015*).

## Construction of transgenic animals

All PCR reactions were performed using PrimeSTAR HS DNA polymerase (Takara #R045Q). All constructs were verified via DNA sequencing (Laragen).

To construct UAS-NPRR$^{ANP}$, *Drosophila* codon-optimized ANP and GCaMP6s were synthesized using gBlocks service (Integrated DNA Technologies), and subcloned into pJFRC7 vector (from Addgene #26220) (*Pfeiffer et al., 2010*) using Gibson cloning. UAS-dTK-NPRR is built in a similar way except the dTK fragment was cloned from the *Drosophila* brain cDNA. NPRR$^{dTK-GFP}$ and NPRR$^{ANP-GFP}$ were built similarly except *Drosophila* codon-optimized GFP was used for the subcloning. All the vectors were injected and integrated into attP2 or attp40 sites (Bestgene Inc; see *Supplementary file 3* for attP sites used for each genotype employed).

## Expression screening of NPRR candidates

Adult fly brains were dissected in chilled PBS and fixed in 4% formaldehyde for 55 min at room temperature. After three 10 min rinses with PBS, the brains were cleared with Vectashield (#1000, Vectorlabs), mounted, and used for native fluorescence measurements. We trace the NPF neuron somata and arborization as ROIs. We selected regions next to NPF neurons and measured its fluorescent intensity as a reference, which represents background autofluorescence. Candidates whose fluorescence reached at least 2-fold higher than reference were selected for functional screening.

## Functional screening of NPRR candidates

For the baseline fluorescence measurement, we crossed NPF-Gal4 to the candidate lines and generated NPF-Gal4>NPRRx (x = candidate label) flies for tests. The dissected adult fly brains were mounted on a petri dish and immersed in *Drosophila* imaging saline (108 mM NaCl, 5 mM KCl, 2 mM CaCl$_2$, 8.2 mM MgCl$_2$, 4 mM NaHCO$_3$, 1 mM NaH$_2$PO$_4$, 5 mM trehalose, 10 mM sucrose, 5 mM HEPES, pH 7.5). To deliver high potassium challenge, High-K imaging saline was perfused (43 mM NaCl, 70 mM KCl, 2 mM CaCl$_2$, 8.2 mM MgCl$_2$, 4 mM NaHCO$_3$, 1 mM NaH$_2$PO$_4$, 5 mM trehalose, 10 mM sucrose, 5 mM HEPES, pH 7.5). Live imaging series were acquired using a Fluoview FV3000 Confocal laser scanning biological microscope (Olympus) with a 40×, 0.8 N.A. (Numerical Aperture) water immersion objective (Olympus). Candidates whose post-stimulation fluorescence reached at least 2-fold of baseline fluorescence (measured as averaged pre-stimulation fluorescence) were selected for in vivo tests at NMJ. For each candidate line, at least three brains were tested and fold-change of each was averaged.

## Immunohistochemistry

Larval dissection was performed in chilled HL3 solution (70 mM NaCl, 5 mM KCl, 20 mM MgCl$_2$, 10 mM NaHCO$_3$, 115 mM sucrose, 5 mM trehalose, 5 mM HEPES and 1.5 mM CaCl$_2$, pH 7.2). Dissected tissues were fixed in 4% formaldehyde or Bouin's solution for 30 min at room temperature. After three 15 min rinses with PBS, tissues were incubated with primary antibodies overnight at 4°C. Following three 15 min rinses with PBS, tissues were incubated with secondary antibody for 2 hr at room temperature. Following three 15 min rinses, tissues were cleared with Vectashield (#1000, Vectorlabs) and mounted. Confocal serial optical sections were acquired using a Fluoview FV3000 Confocal laser scanning biological microscope (Olympus) with a 60×, 1.30 N.A. silicone oil objective (Olympus). All image processing and analyses were done using ImageJ (National Institute of Health).

The following primary antibodies were used: Chicken anti-GFP (1:250-1:1000, Aveslab #1020), Rabbit anti-ANP (1:500, abcam #14348), Guinea pig anti-vGluT (*Goel and Dickman, 2018*) (1:1500), Rabbit anti-syt1 (*Littleton et al., 1993*) (1:500) and Rabbit anti-Bursicon (1:2000, a gift from Dr. Benjamin White).

The following secondary antibodies were used: Alexa Fluor 488 Goat anti-Chicken IgY (#A11039, Invitrogen), Alexa Fluor 488 Goat anti-Rabbit IgG (#A11008, Invitrogen), Alexa Fluor 568 Goat anti-Rabbit IgG(H + L) (#A11011, Invitrogen), Alexa Fluor 633 Goat anti-Rabbit IgG(H + L) (#A21070, Invitrogen), Alexa Fluor 488 Goat anti-Guinea Pig IgG(H + L) (#A11073, Invitrogen), Alexa Fluor 568 Goat anti-Guinea Pig IgG(H + L) (#A11075, Invitrogen), Alexa Fluor 568 Goat anti-Mouse IgG(H + L) (#A11004, Invitrogen) and Alexa Fluor 633 Goat anti-Mouse IgG(H + L) (#A21050, Invitrogen).

## Electron microscopy

*Drosophila* tissues were fixed in 4% formaldehyde in PBS and stored at 4°C until preparation by high-pressure freezing (HPF) and freeze-substitution (FS) (*Buser and Walther, 2008*; *Buser and Drubin, 2013*). Tissues were cryoprotected in 2.3 M sucrose for 45 min, transferred to 200 μm deep planchettes and high-pressure frozen in an EMPact2 with RTS (Leica, Vienna, Austria). FS was carried out in an AFS2 (Leica, Vienna, Austria) in methanol containing 5% water, 0.05% glutaraldehyde and 0.1% uranyl acetate (−90°C, 3 hr; −90 to −80°C, 10 hr; −80°C, 4 hr; −80°C to 4°C, 24 hr). Samples were washed once in methanol containing 5% water, infiltrated with hard grade LR White (Electron Microscopy Sciences, Hatfield, PA, USA) at 4°C ([LR White]: [methanol containing 5% water] 1:1, 24 hr; 100% LR White, 3 × 24 hr) and polymerized in a fresh change of LR White using a Pelco BioWave (Ted Pella, Inc, Redding, CA, USA) set to 750 W, 95°C for 45 min.

60 nm thin sections (UCT ultramicrotome, Leica, Vienna, Austria) were picked up on formvar-coated 50 mesh copper grids. The sections were blocked for 3 min in blocking buffer (PBS with 0.5% bovine serum albumin, which was used for all antibody dilutions), incubated in anti-GFP antibody (1:500, Aveslab #1020) for 5 min, washed 3 times in blocking buffer, incubated in rabbit anti chicken antibody (1:50, MP Biomedicals #55302) for 5 min, washed 3 times on blocking buffer, incubated on protein A - 5 nm gold (1:50, Utrecht, Netherlands), and washed 3 times in PBS and 3 times in distilled water. The sections were stained in uranyl acetate or uranyl acetate and Reynolds lead citrate depending on the desired contrast and imaged at 80 kV in a Zeiss EM10C (Zeiss, Oberkochen, Germany) using a CCD camera (Gatan, Pleasanton, CA, USA).

Labeling density was estimated using stereological methods (*Griffiths and Hoppeler, 1986*). Cross-sections through boutons were recorded and the following parameters were measured: total image area, total number of gold particles, number of visible dense core vesicles (DCV), number of gold particles within a 50 nm radius of the DCV center, bouton area (grid intersection estimate), gold within the bouton cytoplasm, gold within 20 nm of the bouton plasma membrane, gold outside of the bouton (mainly sER). Background labeling was estimated using internal controls (labeling on blank resin and on muscle fibers) and a biological control (non-GFP expressing genotype). Occasional obvious, large gold aggregates were disregarded. Background was consistently below 0.6 gold/μm$^2$ in independently repeated labeling experiments.

## Electrical stimulation

The dissection of third-instar larvae was performed in zero-calcium HL3 saline. The CNS was removed to avoid spontaneous motor neuron activity. To minimize muscle contraction induced by electrical stimulation of motor neurons, the larval body walls were slightly stretched and incubated in HL3 saline supplemented with 10 mM glutamate for 5 mins after dissection to desensitize postsynaptic glutamate receptors. Samples were then shifted to HL3 saline containing 1 mM glutamate and 1.5 mM Ca$^{2+}$. Motor nerves were sucked into a glass micropipette with a stimulation electrode. In *Figure 2* and *Figure 3*, to induce maximum dense core vesicle release at type III motor neuron terminals, four repetitive bursts (70 Hz stimulation for 18–20 s with pulse width of 1 ms) with intervals of 40–42 s were programmed and triggered with a Master-9 stimulator (A.M.P.I., Israel) connected to an iso-flex pulse stimulator (A.M.P.I., Israel). The stimulation intensity was tested and set to double the intensity required to induce muscle contraction by a single pulse stimulation.

In *Figure 4*, stimulation trials were delivered with the same duration, but with a series of frequencies spanning 1 Hz to 70 Hz.

## Calcium imaging

A Nikon A1R confocal microscope with resonant scanner and NIS Element software were used to acquire live Ca$^{2+}$ imaging on third instar larvae, bathed with 1 mM glutamate added in 1.5 mM Ca$^{2+}$

HL3 saline. Type III motor neuron terminals in abdominal segments from A2 to A5 were imaged using a 60x APO 1.4 N.A. water immersion objective with 488 nm excitation laser. A 5 min period was used for time-lapse imaging at a resonance frequency of 1 fps (512 × 512 pixels or 1024 × 1024 pixels), with z-stacks (step length varying from 1 to 1.5 μm) covering the depth of entire type III motor neuron terminals. The repetitive electrical stimulation of 70 Hz was delivered during the imaging session. Samples with severe muscle contractions were abandoned due to imaging difficulties. Maximum intensity projection (MIP) and image registration were conducted using Image J. Plugins including Image Stabilizer (K. Li, CMU) and Template Matching (Q. Tseng) were used for compensating drifting and correcting movement induced by electrical stimulations. ROIs were manually selected by tracing the outer edge of each neuron based on the baseline fluorescence. If the fluorescence was too weak to trace, we established a reference stack by empirically adjusting the contrast on a duplicate of the raw image stack. We used the reference stack for ROI selection and projected the selected ROIs back onto to the raw image stack for measurement. For frames in which the sample movement could not be automatically corrected, we manually outlined the ROIs used for measurements. Preparations with severe movement or deformation artifacts were abandoned to avoid unreliable measurements. Each ROI represent a traceable neuronal branch except *Figure 2—figure supplement 2B*, in which the ROIs were further manually partitioned into boutons and IBIs (Inter-Bouton Intervals) based on morphology. Fluorescence change were normalized to the pre-stimulation background except for *Figure 3E*, for which the data in each trial was normalized to the average $\Delta F/F$ during a 5 s period just before stimulation was initiated. No sample size is predetermined based on statistics. $Ca^{2+}$ imaging data were acquired from at least six independent NMJs from at least five animals.

## Statistical analysis

Data are presented as mean ± s.e.m. All data analysis was performed with Graphpad Prism 6, Microsoft Excel and custom Matlab codes (*Source code 1*). Mann-Whitney U test was used for comparison except in *Figure 4*, where One-sample T test was used for comparison with a specified value (0).

## Acknowledgements

We thank T Südhof for advice and encouragement at the inception of this project, T Littleton for suggesting GCaMP as a potential reporter for DCV release, B Pfeiffer for advice on construct design, members of the Anderson laboratory for critical feedback, K Zinn for helpful comments on the manuscript, P Goel for the information on antibodies, B Kiragasi for Master-9 programming, K Chen and D Ma for advice on imaging analysis and programming, G Mancuso and C Chiu for administrative assistance and lab management, and A Sanchez for maintenance of fly stocks. This work was supported by NIH BRAIN Initiative grant R21EY026432 and R01 DA031389 (to DJA) and NS091546 (to DD). DJA is an Investigator of the Howard Hughes Medical Institute.

## Additional information

### Funding

| Funder | Grant reference number | Author |
| --- | --- | --- |
| National Institutes of Health | R21EY026432 | David J Anderson |
| National Institutes of Health | R01DA031389 | David J Anderson |
| National Institutes of Health | NS091546 | Dion K Dickman |

The funders had no role in study design, data collection and interpretation, or the decision to submit the work for publication.

### Author contributions

Keke Ding, Conceptualization, Data curation, Formal analysis, Investigation, Methodology, Writing—original draft, Writing—review and editing; Yifu Han, Data curation, Formal analysis, Investigation, Methodology, Writing—review and editing; Taylor W Seid, Validation, Investigation, Methodology;

Christopher Buser, Data curation, Formal analysis, Investigation, Methodology; Tomomi Karigo, Investigation, Methodology; Shishuo Zhang, Validation, Investigation; Dion K Dickman, Resources, Formal analysis, Supervision, Funding acquisition, Methodology, Writing—review and editing; David J Anderson, Conceptualization, Data curation, Formal analysis, Supervision, Funding acquisition, Investigation, Methodology, Writing—original draft, Project administration, Writing—review and editing

## Author ORCIDs

Keke Ding https://orcid.org/0000-0002-5261-4843
Yifu Han https://orcid.org/0000-0002-1201-654X
Christopher Buser http://orcid.org/0000-0002-4379-3878
Dion K Dickman http://orcid.org/0000-0003-1884-284X
David J Anderson https://orcid.org/0000-0001-6175-3872

## Decision letter and Author response

Decision letter https://doi.org/10.7554/eLife.46421.sa1
Author response https://doi.org/10.7554/eLife.46421.sa2

## Additional files

### Supplementary files

• Source code 1. Code for data analysis.

• Supplementary file 1. Current techniques for neuropeptide release measurement. Summary of current techniques used for neuropeptide release, including microdialysis, antibody-coated microprobes, GFP-tagged propeptide imaging and NPRR. NA, 'not applicable'

• Supplementary file 2. Stereological labeling estimates. Stereological labeling estimations of $NPRR^{ANP-GFP}$ and $NPRR^{dTK-GFP}$, respectively, in Type Ib neurons, or in Type Ib and Type III neurons. Biological controls and internal controls are described in Materials and methods. SNR: Signal-to-Noise Ratio.

• Supplementary file 3. Complete genotypes of transgenic flies used in this study. Summary of complete genotypes of transgenic flies used in this study.

• Transparent reporting form

### Data availability

Source data of EM for Figure 1 and 3, and codes used for Figure 2 and 3 have been provided.

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
