## [Decision Letter]

Thank you for submitting your article "Imaging neuropeptide release at synapses with a genetically engineered reporter" for consideration by *eLife*. Your article has been reviewed by three peer reviewers, including Leslie C Griffith as the Reviewing Editor and Reviewer #1, and the evaluation has been overseen by Eve Marder as the Senior Editor. The following individual involved in review of your submission has also agreed to reveal their identity: Paul Taghert (Reviewer #3).

The reviewers have discussed the reviews with one another and the Reviewing Editor has drafted this decision to help you prepare a revised submission.

Summary:

This paper develops a new set of tools for looking at release of peptides in real time. The idea is a good one and there is a need in the field for such tools. The authors describe a genetically encoded reporter system for dense core vesicle (DCV) fusion in *Drosophila*. They generate a host of fusions of GCaMP6 to ANF and DTK neuropeptides and identify a specific fusion event that localizes well to DCVs by both immunocytochemistry and immuno-EM. The authors show the sensor can respond to strong stimulation and compare the response to cytosolic GCaMP6 as a read-out of intra-terminal calcium changes. Overall, it's a potentially useful tool for the field, but the study could add a bit more quantification to demonstrate how well the sensor is actually working.

Essential revisions:

1) The major issue the reviewers had with the feasibility of this approach for dissecting DCV release is the conditions needed to trigger these changes – 70 Hz stimulation for 18 secs. These motor neurons would never experience such extreme stimulation in vivo. Peptidergic neurons typically fire in the 1-5 Hz range and while motor neurons fire at higher rates, natural stimulation is not of the duration used here. The signal-to-noise ratio (δ F changes) of this DCV sensor (compared to GCaMP alone which appears to be at least 5-10x more sensitive) is also rather poor. As such, it would be informative to have more information on the dynamics of their new reporter.

a) Provide a dose response curve to demonstrate both the required stimulation frequency (only 70 Hz is reported), the required length of stimulation (only 18 seconds is reported) and the calcium response curve. What does 1 Hz look like? Or 5 Hz?

b) The authors only report changes in the Type 3 terminals. What did the authors see in the Type 1 terminals that have fewer DCVs? Was DCV release detectable in these neurons, as has been reported in the field for the classical ANF-GFP DCV sensor? These data may be more useful than Type III data for investigators contemplating using this tool in neurons that do not have huge numbers of DCVs.

c) It would be helpful to use a benchmark to compare the methodology described here to that currently used in the field – the loss of ANF-GFP signal following stimulation. Is the current system actually an upgrade over that? There is unique value here in the potential to endogenously target specific neuropeptides for this technique, but if the sensitivity is far below that of ANF-GFP, the older system might still be the one of choice for dissecting the biology of DCV fusion overall. Without these data it is hard to claim an advance.

2) It would be helpful for the authors to further clarify in discussion what they actually think the sensor is reporting over time. It was not clear to reviewers exactly what the rise/falling/undershoot/recovery phases actually represent. The recovery phase is especially mysterious since the soma is gone so it cannot be replenishment via transport. DCVs are known to only release part of their cargo during stimulation, so presumably many of these DCVs still contain GCaMP-tether neuropeptide within them. Do the DCVs that have fused become less acidic and take up come extracellular calcium and thus activate the remaining GCaMP within them? Or do the authors think that the DCVs release all their content, or not take up calcium or lose their acidity during partial fusion events? This seems a potential confound – the authors are measuring a transient rise in calcium influx and pH change within the lumen of the DCV, not the actual release of the neuropeptide-GCaMP.

While the reviewers did not feel that it was required that this point be addressed with new experiments, it would be incredibly informative to assay their sensor in animals concurrently expressing an ANF-mCherry, ANF-mOrange (or other red-shifted tagged neuropeptide) where loss of the neuropeptide is directly reported in addition to the neuropeptide-GCaMP signal. That would really help clarify what is likely to be happening with their sensor during these phases of DCV release or partial fusion.

---

## [Author Response]

Essential revisions:
*1) The major issue the reviewers had with the feasibility of this approach for dissecting DCV release is the conditions needed to trigger these changes – 70 Hz stimulation for 18 secs. These motor neurons would never experience such extreme stimulation* in vivo. Peptidergic neurons typically fire in the 1-5 Hz range and while motor neurons fire at higher rates, natural stimulation is not of the duration used here. The signal-to-noise ratio (δ F changes) of this DCV sensor (compared to GCaMP alone which appears to be at least 5-10x more sensitive) is also rather poor. As such, it would be informative to have more information on the dynamics of their new reporter.a) Provide a dose response curve to demonstrate both the required stimulation frequency (only 70 Hz is reported), the required length of stimulation (only 18 seconds is reported) and the calcium response curve. What does 1 Hz look like? Or 5 Hz?b) The authors only report changes in the Type 3 terminals. What did the authors see in the Type 1 terminals that have fewer DCVs? Was DCV release detectable in these neurons, as has been reported in the field for the classical ANF-GFP DCV sensor? These data may be more useful than Type III data for investigators contemplating using this tool in neurons that do not have huge numbers of DCVs.

We share the reviewers’ concern. In response to points 1a and 1b, we have added a new Figure 4, where we present stimulation frequency titrations of NPRR^ANP^, NPRR^dTK^ and GCaMP6 responses in both Type III neurons and Type Ib neurons. Because 70 mM potassium or 70 Hz electric stimulation were reported necessary for inducing maximum dense core vesicle release events in *Drosophila* type III motor neurons ^1^, we covered stimulation frequencies from 1 Hz to 70 Hz as the reviewer requested (*1a*). To compare and contrast the results between preparations and genotypes, for each preparation we normalized the ∆F/F peaks to responses obtained at 70Hz. We did not change the duration of stimulation, as it adds an additional dimension of complexity to the experiments.

Type III motor neurons contain primarily DCVs and exhibit a distinct morphology compared to type Ib motor neurons. As reviewer requested (1b), we have applied the same electric stimulation series to the glutamatergic type Ib motor neurons, which contain both SVs and DCVs ^2^.

To answer the reviewers’ questions in reverse order:

1b) NPRR responses can be observed in Type Ib as well as in Type III neurons (new Figure 4).

1a) In Type Ib neurons, no statistically significant NPRR responses could be observed at 1 Hz (NPRR^ANP^: P=0.2817; NPRR^dTK^: P=0.0723) or 5 Hz (NPRR^ANP^: P=0.0727; NPRR^dTK^: P=0.3467). However, responses at 10 Hz were statistically significant (NPRR^ANP^: P=0.0108; NPRR^dTK^: P=0.0124). GCaMP responses were statistically significant at 5 Hz (P=0.0152) but not at 1 Hz (P=0.1194). Importantly, unlike the case in Type III neurons, we did not observe significant differences in the performance of the NPRR^ANP^ and NPRR^dTK^ reporters in Type Ib neurons (Figure 4—figure supplement 1). In Type III neurons, no statistically significant responses were observed at 1 Hz (NPRR^ANP^: P=0.3304; NPRR^dTK^: P=0.0719), 5 Hz (NPRR^ANP^: P=0.3699; NPRR^dTK^: P=0.3467) and 10 Hz (NPRR^ANP^: P=0.3084; NPRR^dTK^: P=0.5660). We observed a statistically significant increase in ∆F/F at 30 Hz for NPRR^ANP^ but not NPRR^dTK^; at 50 Hz and above we observed significant responses with both reporters. (NPRR^ANP^: P=0.0021; NPRR^dTK^: P=0.034). The difference between both NPRRs at 30 Hz or 50 Hz was also statistically significant ((Figure 4—figure supplement 1).

(*Statistical methods: For in-sample comparison, one-sample T-test was used to compare response at a given stimulation frequency to 0. For between-sample comparison at the same stimulation frequency, Mann-Whitney U test was used.)

These results reveal two important new findings: (1) the difference between the performance of the two NPRRs we tested in Type III neurons is not a general property of these reporters, but a peculiarity of Type III neurons; (2) the high threshold for detection of NPRR responses in Type III neurons (30-50 Hz) is also not a general property of these reporters, since significant responses can be observed at frequencies as low as 10 Hz in Type 1b neurons. 10 Hz is well within the range of physiological rates for many neurons. These observations should allay concerns that NPRRs will have limited applicability.

Indeed, these new results reveal a novel application for NPRRs: they can be used to compare the relationship between stimulation and neuropeptide release between different types of neurons. In the present case, the difference in NPRR release detection thresholds between Type Ib and Type III neurons appears to be correlated with differences in the overall responses of these two cell types to electrical stimulation, as revealed by conventional GcaMP imaging (Figure 4C, F). However further studies will be necessary to substantiate this. To our knowledge, such a comparison of NP release thresholds between different neuronal cell types has not been performed previously. These data therefore illustrate a new area of investigation that is opened up by this technology.

In response to the reviewer’s comment that the signal-to-noise ratio of this sensor is rather poor, in comparison to that of GCaMP6, we would respectfully point out that the signal-to-noise ratio of the first-generation GCaMPs were also rather poor (max ∆F/F ≪100%). It took a managed team of protein engineers at Janelia 5 years and $10 million to optimize GCaMP from version 1.3, to version 6 (the first version with single-spike sensitivity). We would respectfully submit that the expectation that a first-generation reporter of a new class would perform as well as the 6th generation GcaMP is unrealistic. We anticipate that there will be multiple iterations of NPRRs to optimize their performance, but such optimization is beyond the scope of this manuscript.

c) It would be helpful to use a benchmark to compare the methodology described here to that currently used in the field – the loss of ANF-GFP signal following stimulation. Is the current system actually an upgrade over that? There is unique value here in the potential to endogenously target specific neuropeptides for this technique, but if the sensitivity is far below that of ANF-GFP, the older system might still be the one of choice for dissecting the biology of DCV fusion overall. Without these data it is hard to claim an advance.

We took the reviewers’ point and attempted to benchmark our method to ANP-GFP. While both methods showed comparable decreases in fluorescence in response to 70 mM KCl (see below), in response to electrical stimulation, we failed to observe any decrease in ANP-GFP signals in Type Ib or Type III neurons (Author response image 1A_1,2_). In contrast, under the same stimulation conditions NPRR^ANP^ clearly exhibited an increase in dF/F (Author response image 1A_3,4_). In other words, in our hands the ANPGFP method yielded no signal (decreased fluorescence) in response to electrical stimulation, whereas the NPRR clearly did so. Therefore rather than it being the case that the sensitivity of NPRRs is “far below that of ANF-GFP,” it appears that NPRRs are far more sensitive than ANF-GFP, in the context of electrical stimulation.

**Author response image 1. respfig1:** Benchmarking NPRR with ANP-GFP. Trace from a representative experiment showing changes in fluorescence intensity (∆F/F) of ANP-GFP and NPRR^ANP^ evoked by electrical stimulation (Red bar, frequency shown in A1). No significant ANP-GFP fluorescent change was observed within stimulations, in both Type Ib (A1) and Type III neurons (A2), while NPRR^ANP^ displays distinct characteristics in Type Ib (A3) and Type III(A4). Green line: mean ∆F/F; Grey shading: s.e.m envelope.

Details: We measured the native fluorescent intensity of ANP-GFP or of NPRR^ANP^ directly in fixed specimens following stimulation with 70 mM KCl. In this case, we observed a significant decrease in ANP-GFP fluorescence (36.04% decrease, N=7) after 10 mins, consistent with published work from the Levitan laboratory ^1,3^. In parallel, NPRR^ANP^ underwent a comparable level of fluorescent loss (33.57% decrease, N=7), suggesting a similar sensitivity of the two reporters under these conditions of bath depolarization.

We cannot explain our failure to observe any fluorescent loss of ANP-GFP in response to tenable electrical stimulation, in contrast to published results from the Levitan laboratory ^1,3^. Whatever the explanation, our results demonstrate the superiority of the NPRR methodology. In addition to being able to visualize specific neuropeptides, the ultimate objective of generating NPRRs is to be able to measure NP release in behaving animals over time. For this purpose, a positive signal would be more optimal than a negative signal (ANP-GFP), as fluorescent loss could be confounded by other technical issues, such as photobleaching or movement.

2) It would be helpful for the authors to further clarify in discussion what they actually think the sensor is reporting over time. It was not clear to reviewers exactly what the rise/falling/undershoot/recovery phases actually represent. The recovery phase is especially mysterious since the soma is gone so it cannot be replenishment via transport. DCVs are known to only release part of their cargo during stimulation, so presumably many of these DCVs still contain GCaMP-tether neuropeptide within them. Do the DCVs that have fused become less acidic and take up come extracellular calcium and thus activate the remaining GCaMP within them? Or do the authors think that the DCVs release all their content, or not take up calcium or lose their acidity during partial fusion events? This seems a potential confound – the authors are measuring a transient rise in calcium influx and pH change within the lumen of the DCV, not the actual release of the neuropeptide-GCaMP.

We appreciate the reviewers’ request to clarify the temporal kinetics we observed. As we stated in the original submission, we interpret the rising phase as reflecting epochs of NP release. The falling phase where an undershoot was observed suggests that the overall decrease of GCaMP fluorescence represents a reduction of total DCV NPRR cargo in the bouton area following vesicle fusion.

The reviewers state that the recovery phase is mysterious, because replenishment via transport cannot occur in the absence of the soma, which is removed in these preparations. The reviewer’s statement is partially incorrect: recent studies have demonstrated that axonal transport indeed occurs during short-term presynaptic protein remodelling with^4^ or without^5^ the ventral verve cord (i.e., somata) removed, over a time course ranging from 10 – 30 min ^6,7^, a timeframe similar to that in our experimental design. Thus, replenishment of DCVs via continuous transport from unstimulated axon segments contained within our preparation seems the most parsimonious explanation for the recovery phase.

DCV release occurs in several modes. At this point, we are unable to determine whether an individual DCV unloads all or a portion of its content in our experiments, although as mentioned above (response to point 1c) we detected an ~36% decrease in overall GCaMP fluorescence following high potassium stimulation. Furthermore, the ∆F/F undershoot observed in the recovery phase supports the idea that at least some, if not all, NPRR molecules are released to the extracellular space, since a purely intravesicular change in calcium/pH would not cause such an undershoot. Nevertheless, we cannot fully exclude the possibility that some fraction of NPRR increased ∆F/F signal is contributed by unreleased NPRR molecules, and now acknowledge this in the Discussion (second paragraph).

While the reviewers did not feel that it was required that this point be addressed with new experiments, it would be incredibly informative to assay their sensor in animals concurrently expressing an ANF-mCherry, ANF-mOrange (or other red-shifted tagged neuropeptide) where loss of the neuropeptide is directly reported in addition to the neuropeptide-GCaMP signal. That would really help clarify what is likely to be happening with their sensor during these phases of DCV release or partial fusion.

We agree with the reviewers’ suggestion to have an additional channel to report neuropeptide loss directly. In fact, we initially generated sixteen different neuropeptide precursor-Red Fluorescent Protein fusions (Figure 1—figure supplement 1). Unfortunately, all 16 red NPRRs exhibited 1) nearly invisible baseline fluorescence; and 2) no fluorescence change following high potassium challenge (unlike GFP fusions). We tried to investigate the reason for this failure using Neuropred (http://stagbeetle.animal.uiuc.edu/cgi-bin/neuropred.py), an online algorithm that predicts potential peptidergic cleavage sites within a given protein. This analysis revealed that RFP and tdTomato have 10 and 25 predicted cleavage sites, respectively, as opposed to 3 sites in GCaMP6s, mainly reflecting the high abundance of positively-charged residue repeats (Lys-Lys, Lys-Arg) in RFPs. If these cleavage sites are accessible to DCV proteases that processes packaged NP precursors, it would explain why no red fluorescence was observed with these reporters. We have now mentioned this in the text. Apparently, generating such RFP-based reporters would require mutagenizing all such sites to avoid cleavage, while retaining red fluorescence. Such protein engineering is desirable, but outside the scope of the present manuscript.

We appreciate the suggestion to use orange fluorescent proteins such as mOrange or mKO as a fusion alternative. Although it was not included in our first round of design, we would be glad to try it in the future.

References:

1) Levitan, E. S., Lanni, F. and Shakiryanova, D. In vivo imaging of vesicle motion and release at the Drosophila neuromuscular junction. Nat Protoc 2, 1117-1125, doi:10.1038/nprot.2007.142 (2007).

2) Rao, S., Lang, C., Levitan, E. S. and Deitcher, D. L. Visualization of neuropeptide expression, transport, and exocytosis in Drosophila melanogaster. J Neurobiol 49, 159-172 (2001).

3) Shakiryanova, D., Tully, A. and Levitan, E. S. Activity-dependent synaptic capture of transiting peptidergic vesicles. Nat Neurosci 9, 896-900, doi:10.1038/nn1719 (2006).

4) Goel, P. et al. Homeostatic scaling of active zone scaffolds maintains global synaptic strength. J Cell Biol 218, 1706-1724, doi:10.1083/jcb.201807165 (2019).

5) Bohme, M. A. et al. Rapid active zone remodeling consolidates presynaptic potentiation. Nat Commun 10, 1085, doi:10.1038/s41467-019-08977-6 (2019).